# Predictors of Short-Term Alcohol Drinking in Patients with Alcohol Use Disorders during the Third Wave of the COVID-19 Pandemic: Prospective Study in Three Addiction Outpatient Centers in France

**DOI:** 10.3390/ijerph19041948

**Published:** 2022-02-10

**Authors:** Aymery Constant, Marlène Sanz, Romain Moirand

**Affiliations:** 1Department of Behavioral Sciences, EHESP School of Public Health, 35043 Rennes, France; 2INRAE, INSERM, Nutrition Metabolisms and Cancer Institute, University of Rennes1, 35000 Rennes, France; romain.moirand@univ-rennes1.fr; 3CH de Saint Malo, Service D’addictologie, 35400 Saint Malo, France; golumaaa@hotmail.fr; 4CHU de Rennes, Unité D’addictologie, 35000 Rennes, France

**Keywords:** craving, alcohol use disorders, alcohol drinking, prospective study, COVID-19, France

## Abstract

The present study investigates the extent to which the COVID-19 crisis disturbed different life domains of patients with alcohol use disorder (AUD) and assessed the associations between these disturbances and the risk of short-term alcohol drinking. All patients aged >18 years receiving outpatient care at three addiction treatment facilities from 15 April to 30 May 2021 were eligible for inclusion in the study. A trained resident assessed the extent to which the COVID-19 crisis affected their professional activity, social life, access to healthcare, and drinking problems, together with craving, drinking behavior, psychological distress, physical/mental health, and sociodemographic and clinical data. The same investigator assessed alcohol drinking 1 month after their visit. Nearly half of the patients felt that the COVID-19 crisis had a serious impact on their drinking problems, despite minor disruptions in access to healthcare. These disturbances significantly influenced short-term alcohol drinking in univariate analysis, together with psychological distress, craving, and drinking problems. Only craving predicted alcohol drinking in multivariate analyses, suggesting that psychological and drinking problems, as well as COVID-19 disturbances, increased the risk of alcohol drinking by increasing craving. Craving should be systematically investigated in patients with AUD to establish adapted social support systems during pandemics.

## 1. Introduction

The coronavirus (COVID-19) pandemic has had a global impact, with more than ten million cases and more than 500,000 deaths by 1 July 2020 [1]. Several measures were implemented to prevent further spread of the disease in the early stages of the pandemic. Lockdown, the restriction of individuals to their homes, was one of the measures enforced in many countries, including France. Lockdown and physical distancing helped control the first wave of the COVID-19 pandemic, but also had significant negative consequences for individuals’ mental and physical health [2,3], especially those with high levels of COVID-19 anxiety [4]. Over the following 18-month period, stay-at-home restrictions were extended, leading to prolonged isolation, boredom, and depression [5]. The possible impact of the negative side effects of preventive measures on addictive behaviors emerged rapidly as a major public health problem in the midst of quarantine [6]. 

In particular, alcohol-related problems constituted a major health concern, as studies indicated that harmful drinking increased during the lockdown among some individuals as a coping response to COVID-19 distress and social isolation [7,8,9]. In addition, the stay-at-home order for non-essential workers and mobility restrictions reduced the workforce across all economic sectors and caused many jobs to be lost [10]. These socio-economic consequences of the pandemic jeopardized social and professional activities in whole segments of the population, especially in the most deprived subgroups, who bear a disproportionate burden of the negative alcohol-related consequences [11]. Finally, the pandemic has resulted in the disruption of a range of services, including emergency, treatment, and relapse prevention and liaison services [12,13,14] during a period of time in which mental health needs increased dramatically [15]. Altogether, a complex interaction of financial, social, psychological, and professional difficulties combined with the upheaval of clinical services could have contributed to complications in patients with pre-existing alcohol use disorder (AUD) during the COVID-19 pandemic, such as increased alcohol consumption for active drinkers and relapse for those who were previously abstinent [16,17].

A better understanding of the factors underlying vulnerability to alcohol drinking during the COVID-19 crisis may contribute to designing more effective public health programs and services addressing AUD in the current and future pandemics. The objectives of the present study were to investigate the extent to which the COVID-19 crisis disturbed different life domains (social and professional lives, access to healthcare, and drinking problems) of patients with AUD and to assess the associations between these disturbances and the risk of short-term alcohol drinking.

## 2. Materials and Methods

We conducted a prospective study in patients with AUD during the third wave of the COVID-19 pandemic in a region where alcohol consumption and the incidence of problems are particularly high. All patients aged >18 years meeting DSM-5 criteria for AUD who were receiving outpatient care at three addiction treatment facilities in the same geographic area of Britany (France) from 15 April to 30 May 2021 were eligible for inclusion in the study. These medical facilities included staff from different backgrounds to provide support and healthcare to patients with AUD, including medical doctors, nurses, social workers, and neuropsychologists. The exclusion criteria were current or recent (<4 days) alcohol consumption, high alcohol intoxication, significant cognitive impairment or severe physical or mental illness, incompetence in terms of providing informed consent, non-fluency in French, major legal protection (safeguarding justice, guardianship, or trusteeship), acute manic phase or psychotic decompensation, confusion state, and refusal. The Institutional Review Board (IRB) of the Rennes University Hospital approved the study protocol (IRB number: 21–181), and all participants provided informed consent. 

### 2.1. Procedure and Baseline Assessment

A trained addiction psychiatrist (MS) interviewed the patients upon their arrival at the addiction centers for informed consent and the baseline assessment. The same investigator had a phone interview with each participant 28 days after the baseline assessment to assess alcohol drinking, which was defined as any alcohol consumption since baseline. The following data were collected at baseline: age (in years), sex, marital status (living alone or with a partner), diploma (no high school degree, high school degree/further education), professional activity (active, unemployed, retired), current medical treatment for mental health (yes/no) and craving (yes/no), and history of mood disorders (yes/no).

### 2.2. Measures

The Alcohol Urge Questionnaire (AUQ) is an eight-item self-reported questionnaire that assesses drinking urges [18] and has been validated in French [19]. Questions are in the form of a 7-point Likert scale, and participants endorse the extent to which they agree or disagree with statements relating to their desire to drink, expectation of a desired outcome from drinking, and inability to avoid drinking if alcohol is available.

The Alcohol Use Disorders Identification Test (AUDIT) is a 10-item screening tool developed by the World Health Organization (WHO) to assess alcohol consumption, drinking behaviors, and alcohol-related problems [20]. It also provides a framework for interventions to help risky drinkers reduce or cease alcohol consumption. A score ≥8 is considered to indicate possible hazardous or harmful alcohol use. A score ≥13 indicates possible dependence. The AUDIT has been validated across genders and in a wide range of racial/ethnic groups and languages, including French [21], and is well suited for use in primary care settings.

Psychological distress was assessed using the French version of the Hospital Anxiety and Depression Scale (HADS) [22,23], which is widely used to measure psychological morbidity in patients with a variety of medical and psychological conditions. The scale consists of 14 items: 7 assess anxiety symptoms (HADS-A), and 7 assess depression symptoms (HADS-D). Each item is graded from 0 (not present) to 3 (maximum), and the total score ranges from 0 (no symptoms) to 21 (high symptoms).

Physical and psychological functioning were assessed using the French version of the SF-36 [24], a multipurpose, 36-item survey that measures and scores eight domains of health-related quality of life: physical functioning, role limitations due to physical health, bodily pain, general health perceptions, vitality, social functioning, role limitations due to emotional problems, and mental health. A lower score indicates poorer health-related quality of life. These eight scales can be aggregated into two summary measures, the physical component score (PCS) and the mental health component score (MCS) [25].

Participants estimated the extent to which the COVID-19 crisis affected their professional activity (for employed participants), social life (e.g., social activities, meeting with friends/family, etc.), access to healthcare (e.g., general practitioners, hospitals, or pharmacy), and drinking problems using a 5-point scale for each domain (none = 1, minor = 2, moderate = 3, high = 4, severe = 5).

### 2.3. Statistical Analysis

Categorical data are expressed as numbers and percentages, whereas numerical data are expressed as means ± standard deviations (SDs) or as medians and interquartile ranges (IQRs). As our study outcome was a binary variable (alcohol drinking: yes/no), we used a logistic regression model to estimate the odds ratios (ORs) for alcohol drinking 28 days after baseline as a function of COVID-19 disturbances together with sociodemographic and psychological variables assessed at baseline. Estimates (model 1) are expressed as the OR with 95% confidence interval (CI). Significant estimates from model 1 were analyzed in a multivariate model (i.e., model 2). Statistical analyses were performed using the SPSS statistical package, version 19 (SPSS, Chicago, IL, USA).

## 3. Results

A total of 80 patients receiving outpatient care in the three participating addiction centers (CSAPA Saint Malo, 47.5%; CSAPA Dinan, 38.8%; Day Hospital, 13.8%) agreed to participate in the study (Table 1).

The mean age was 50.3 ± 10.8 years, with the majority of participants being male (60.0%), living alone (62.5%), and professionally active (70.0%), and a minority had a high school degree or further education (48.7%). All but two patients presented with severe AUD (median number of DSM5 criteria: 9 (IQR 8–11)), and 63.8% had a history of mood disorders. Although all participants were sober at inclusion, the period of abstinence varied across the study sample, ranging from 4 to 970 days (median number of days without alcohol: 19.5 (7–116)). Regarding psychological variables, patients reported AUDIT scores above the cut-off values, indicating severe AUD, and the SF-36 physical component was less than the standardized mean score of 50. However, they also reported low scores for anxiety (8.7 ± 4.6), depressive symptoms (3.7 ± 3.3), and craving (AUQ: 13.9 ± 6.4), and the SF-36 mental health component was higher than the standardized mean score of 50. In addition, the majority of patients received treatment for mental health problems (80.0%) and craving (57%). Follow-up assessment revealed that alcohol drinking occurred in 43.2% of participants in the 28-day follow-up period.

The COVID-19 pandemic highly or severely affected drinking problems for 43.8% of patients and social life for 47.6% of patients (Table 2). 

In addition, 34.7% of employed participants (N = 56) reported a high or severe impact on their professional activity. Conversely, the COVID-19 pandemic highly or severely affected access to healthcare for fewer than 1 in 5 patients (17.6%).

In the univariate analysis (Table 3, model 1), the risk of alcohol drinking increased significantly in patients with higher education status (OR = 2.70 (95% CI 1.12–6.94)) and more anxiety (1.16 (1.04–1.30)) and depressive symptoms (1.26 (1.08–1.47)), along with greater craving (2.15 (1.48–3.10)), severity of drinking problems (1.08 (1.01–1.15)), and disturbances in drinking problems due to COVID-19 (1.63 (1.16–2.30)). The risk of alcohol drinking decreased with a higher SF-36 MCS (0.94 (0.90–0.97)) and a greater number of days since the last drink (0.92 (0.87–0.97)). In the multivariate analysis, the risk of alcohol drinking increased significantly with craving (2.97 (1.16–7.60)).

## 4. Discussion

Nearly half of the patients receiving outpatient care for AUD felt that the COVID-19 crisis had a serious impact on their drinking problems, despite minor disruptions in access to healthcare. These disturbances significantly influenced short-term alcohol drinking in the univariate analysis, together with psychological distress, craving, and drinking problems. By contrast, better psychological functioning and a longer abstinence period at inclusion decreased the risk of alcohol drinking. Only craving predicted alcohol drinking in the multivariate analysis, suggesting that psychological and drinking problems, as well as COVID-19 disturbances, increased the risk of alcohol drinking by increasing craving.

Regarding social characteristics, such as male gender, education status, living alone, or psychiatric co-morbidities, our participants were representative of patients hospitalized for alcohol issues in France, whose characteristics have been investigated previously [26]. Our participants reported below-average physical functioning, as alcohol dependence includes impairment of functional status [27,28], but they also reported above-average psychological functioning and low levels of alcohol urges and psychological distress. These symptoms, which are prevalent in individuals with alcohol dependence [29], seemed to be under control in our study population, probably because of the mental healthcare they received in outpatient centers.

However, more than 4 in 10 participants reported alcohol drinking in the month following their inclusion, which is a higher rate than that reported in other studies conducted during the COVID-19 pandemic [17,30], and periods of abstinence varied considerably from patient to patient. One possible explanation is that our participants were in outpatient treatment, which allows the maintenance of day-to-day activities [31] but exposes patients to drugs or alcohol. Moreover, data were collected during the third wave of the COVID-19 pandemic in France, when confinement and mobility restrictions were implemented for more than a year, along with related disturbances and psychological side effects.

This long-lasting crisis provoked significant disturbances in social and professional activities, increasing loneliness and isolation in many segments of the population [32,33,34]. In the early stages of the pandemic, authorities closed schools, universities, and non-essential businesses, and the need for commodities and manufactured products has decreased [10]. Consequently, more than 1 in 3 patients reported that the COVID-19 pandemic significantly disturbed their social and professional lives. This is even more concerning as patients with AUD tend to be isolated [35], and most of the participants in the current study lived alone. By contrast, most of the participants reported minor disturbances in access to healthcare, despite the considerable upheaval in the healthcare system. Notably, the participating treatment centers implemented telephone consultation services in the early stages of the pandemic for patients who needed emergency help, whereas inpatient care remained available in case of relapse. These adaptations in service delivery, together with the rapid development of teleconsultations, may have partly contributed to the relative continuation of healthcare during the COVID-19 crisis [36]. However, this was insufficient to tackle all the consequences of the COVID-19 pandemic, as 4 in 10 patients reported significant disturbances in their drinking problems. This is in line with data showing that individuals reported increased alcohol intake to cope with emotional stress and chronic uncertainty during the stay-at-home restrictions [37], which may have exacerbated alcohol-related problems in individuals with pre-existing addiction disorders [17,30,38]. Therefore, the COVID-19 pandemic is added to the list of stressful life events that may jeopardize mental health outcomes in patients with AUD [39].

The relationships observed between COVID-19 disturbances, the severity of alcohol problems, and other indicators of psychological distress became non-significant when combined with craving in the multivariate model. In addition, the risk of alcohol drinking was lower in patients with a longer period of abstinence before inclusion in the univariate analysis, as craving and other negative effects tended to decline over time after patients initiated abstinence [40], but this relationship became marginally non-significant in the multivariate model. Taken together, these findings indicate that COVID-19 disturbances and psychological distress influenced the risk of alcohol drinking by increasing craving, a result that confirms that alcohol urge is the main predictor of relapse when patients with AUD have to deal with difficult situations.

The picture was different regarding the influence of participants’ sociodemographic characteristics on alcohol drinking. Living alone is associated with a greater risk of suffering from AUD, whereas social factors and support play an important role in preventing relapse [41,42]. Many participants in the present study were unemployed, lived alone, and had a low education status and a history of mood disorders, but these indicators of social vulnerability were unrelated to the risk of alcohol drinking. It is possible that social support and activities provided to patients in the addiction centers somewhat attenuated the consequences of the consecutive stay-at-home orders [37]. Notably, the risk of alcohol drinking was similar between the three participating centers, suggesting coherent healthcare practices throughout these local institutions.

This study has several limitations. First, the study was conducted at addiction centers in a specific region of France where the incidence of alcohol problems is particularly high, which may limit the generalizability of the results. The sample population has the advantage of being under similar alcohol management and medical treatment, and most participants were detoxified at baseline, though with important variations in their sobriety periods. Second, highly intoxicated patients were excluded from the study because of their inability to understand the study objectives and provide informed consent. Third, the relatively small size of the sample results in a lack of power when exploring small correlations. Finally, how the COVID-19 crisis could have positively affected their drinking problems was not assessed, although the limitation of social interactions could also have limited temptations to drink alcohol during the COVID-19 pandemic.

## 5. Conclusions

A large proportion of patients receiving outpatient care for AUD reported that the COVID-19 pandemic affected their drinking problems. These disturbances, together with psychological distress, influenced the risk of short-term alcohol drinking through increasing craving. This mediating effect confirms that craving remains the main predictor of alcohol drinking when patients are in difficult situations and should be systematically investigated to establish adapted social support systems during pandemics.

## Figures and Tables

**Table 1 ijerph-19-01948-t001:** Characteristics of participants receiving outpatient care for alcohol use disorders in the three participating addiction centers (N = 80).

Variable		N (%) or Mean (SD)
Male sex		48 (60.0%)
Age, years		50.3 (10.8)
Living with partner		30 (37.5%)
High school degree or more		39 (48.7%)
Professional status	Active	56 (70.0%)
	Unemployed	11 (13.7%)
	Retired	13 (16.3%)
Addiction treatment center	CSAPA Dinan	31 (38.8%)
	CSAPA Saint Malo	38 (47.5%)
	Day Hospital	11 (13.8%)
Current medical treatment	Mental health	64 (80.0%)
	Craving	46 (57.0%)
History of mood disorders		51 (63.8%)
Median number of days since the last drink	Range: 4–970	19
AUDIT score	Range: 0–40	28.8 (7.2)
HADS Anxiety score	Range: 0–21	8.7 (4.6)
HADS Depression score	Range: 0–21	3.7 (3.3)
AUQ score	Range: 8–56	13.9 (6.4)
SF-36 Physical score	Range: 0–100	49.1 (8.0)
SF-36 Mental score	Range: 0–100	56.6 (14.9)
Alcohol drinking at 28 days		35 (43.8%)

SD = standard deviation; AUDIT = Alcohol Use Disorders Identification Test; HADS = Hospital Anxiety and Depression Scale; AUQ = Alcohol Urge Questionnaire.

**Table 2 ijerph-19-01948-t002:** Reported impact of COVID-19 on four life domains in patients with alcohol use disorder.

Life Domains	Mean Score (SD)	N (%)				
		None	Minor	Moderate	High	Severe
Professional activity ^†^	3.0 (1.6)	15 (26.8)	8 (14.3)	8 (14.3)	9 (16.1)	16 (18.6)
Social life	3.3 (1.4)	11 (13.8)	14 (17.5)	17 (21.3)	17 (21.3)	21 (26.3)
Access to healthcare	2.1 (1.3)	38 (47.5)	13 (16.3)	15 (18.8)	7 (8.8)	7 (8.8)
Drinking problems	3.1 (1.5)	17 (21.3)	12 (15.0)	16 (20.0)	17 (21.3)	18 (22.5)

SD = standard deviation; ^†^ Assessed in professional active participants (N = 56).

**Table 3 ijerph-19-01948-t003:** Determinants of short-term alcohol drinking in patients with alcohol use disorder (N = 80) in the 28-day period after baseline.

Variables		Model 1	Model 2
		Univariate OR	Multivariate Model
Male sex		0.43 (0.17–1.07)	
Age, years		0.99 (0.95–1.03)	
Living with partner		1.03 (0.41–2.56)	
High school degree		**2.79 (1.12–6.94)**	0.75 (0.06–9.00)
Professionally active		0.89 (0.34–2.32)	
Outpatient Addiction center	CSAPA Saint-Malo	1.77 (0.44–7.00)	
	CSAPA Dinan	1.10 (0.27–4.60)	
	Day Hospital	1	
Current medical treatment	Mental health	1.38 (0.45–4.25)	
	Craving	0.97 (0.40–2.38)	
History of mood disorders		1.83 (0.71–4.70)	
Number of days since the last drink		**0.92 (0.87–0.97)**	0.91 (0.82–1.01)
Drinking problems, AUDIT		**1.08 (1.01–1.15)**	0.98 (0.80–1.21)
HADS score	Anxiety	**1.16 (1.04–1.30)**	1.25 (0.91–1.71)
	Depression	**1.26 (1.08–1.47)**	0.63 (0.36–1.11)
Craving, AUQ		**2.15 (1.48–3.10)**	**2.97 (1.16–7.60)**
SF-36 component score	Physical	1.02 (0.97–1.08)	
	Mental	**0.94 (0.90–0.97)**	0.95 (0.86–1.06)
Disturbances by the COVID-19 crisis	Professional activity	1.09 (0.83–1.44)	
	Social life	1.11 (0.80–1.53)	
	Healthcare access	1.11 (0.80–1.55)	
	Drinking problems	**1.63 (1.16–2.30)**	1.50 (0.52–4.29)

Odds ratios (ORs) and 95% confidence intervals (CIs) were estimated using logistic regression. Values are presented as OR (95% CI). SD = standard deviation; AUDIT = Alcohol Use Disorders Identification Test; HADS = Hospital Anxiety and Depression Scale; AUQ = Alcohol Urge Questionnaire. Significant results (*p* < 0.05) are in bold.

## Data Availability

The dataset generated and analyzed during the current study is available. Open Science Framework. Available online: https://osf.io/gdksf (accessed on 29 January 2022).

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
