# Peer review of "Predictors of Short-Term Alcohol Drinking in Patients with Alcohol Use Disorders during the Third Wave of the COVID-19 Pandemic: Prospective Study in Three Addiction Outpatient Centers in France"

_ijerph, 2022, doi:10.3390/ijerph19041948_

Round 1

Reviewer 1 Report

The authors present an observational study to investigate 12 life domains of patients with alcohol use disorder (AUD). These domains along with the eventual relapse were assessed through questionnaires. Nearly half of the patients felt that the COVID-19 crisis had a serious impact on their drinking problems, despite minor disruptions in access to healthcare. Craving predicted relapse in multivariate analyses, suggesting that psychological and dinking problems, as well as COVID-19 disturbances, increased the risk of relapse by increasing craving. 

The study is well constructed although relapse should have been confirmed also by alcohol markers such as CDT, EtG in urine or PEth in blood.

The results are clearly presented and are of interest for the scientific community 

Author Response

Dear Sir/Ma,

Many thanks for taking the time to review our manuscript. We have addressed the comments and revised the manuscript accordingly. We found the comments helpful and believe that our revised manuscript represents a significant improvement over our initial submission.

We hope the revised version would be suitable for publication and look forward to your feedback in due course.

Thank you

Reviewer 2 Report

This is an interesting study on patients with AUD during 3 wave of COVID-19 pandemic in France. However, the improve the quality of the manuscript the authors are suggested to do the following updates,

  1. Include the AUD patients with COVID-19 infection rate in France.
  2. What  are the other complications that chronic AUD patients faced during third wave of COVID-19.
  3. What are many home made remedy to resolve the initial AUD patients, include that points.

Author Response

We are grateful for the helpful comments and suggestions. Questions and concerns noted by the reviewer are addressed below.

Comment #1: Include the AUD patients with COVID-19 infection rate in France.

Answer#1: To the best of our knowledge, there is no estimate of COVID-19 infection are in French patients with AUD. We found only one study linking alcohol consumption and risk of contracting the COVID-19, but results are mixed, the relationship is unclear and probably influenced by socio-economic confounders

Comment #2: What are the other complications that chronic AUD patients faced during third wave of COVID-19.

Answer#2: Apart from social isolation and treatment discontinuation, we found no other complications specific of AUD patients in France.

Comment #3: What are many home made remedy to resolve the initial AUD patients, include that points

Answer#3: Home remedies to leave alcohol addiction are absent from the French culture, to the best of our knowledge. Healthcare services and general practioners are the main resources for quitting Alcohol, although denial is prominent in these patients.